# Contamination identification, source apportionment and health risk assessment of trace elements at different fractions of atmospheric particles at iron and steelmaking areas in China

Xiaoteng Zhou[1,2]*, Vladimir Strezov[1,2], Yijiao Jiang[1,3], Xiaoxia Yang[1,3], Tao Kan[1,2], Tim Evans[1,2]

1 ARC Research Hub for Computational Particle Technology, Macquarie University, Sydney, New South Wales, Australia, 2 Department of Earth and Environmental Sciences, Macquarie University, Sydney, New South Wales, Australia, 3 School of Engineering, Macquarie University, Sydney, New South Wales, Australia

* xiaoteng.zhou@mq.edu.au

**Data Availability Statement:** All relevant data are within the manuscript and its Supporting Information files.

## Abstract

China has the largest share of global iron and steel production, which is considered to play a significant contribution to air pollution. This study aims to investigate trace element contamination at different fractions of particulate matter (PM) at industrial areas in China. Three PM fractions, $PM_{2.1-9.0}$, $PM_{1.1-2.1}$ and $PM_{1.1}$, were collected from areas surrounding iron and steelmaking plants at Kunming, Wuhan, Nanjing and Ningbo in China. Multiple trace elements and their bioavailability, as well as Pb isotopic compositions, were analysed for identification of contaminants, health risk assessment and source apportionment. Results showed that PM particles in the sites near industrial areas were associated with a range of toxic trace elements, specifically As, Cr(VI), Cd and Mn, and posed significant health risks to humans. The isotopic Pb compositions identified that coal and high temperature metallurgical processes in the steelmaking process were the dominant contributors to local air pollution in these sites. In addition to iron and steelmaking activities, traffic emissions and remote pollution also played a contributing role in PM contamination, confirmed by the differences of Pb isotopic compositions at each PM fraction and statistical results from Preference Ranking Organization Method for Enrichment Evaluations (PROMETHEE) and Geometrical Analysis for Interactive Aid (GAIA). The results presented in this study provide a comprehensive understanding of PM emissions at iron and steelmaking areas, which helps to guide subsequent updates of air pollution control guidelines to efficiently minimise environmental footprint and ensure long term sustainability of the industries.

**Funding:** This project is founded by the Australian Research Council Industrial Transformation Research Hub for Computational Particle Technology (IH140100035), Australia.

**Competing interests:** The authors have declared that no competing interests exist.

## Introduction

Iron and steel manufacturing is a staple of the world's industrial economy with a total estimated worth of $900 billion per year [1]. Almost everything used today, such as housing, transport, energy production, food and water supply, is either made of steel or manufactured by steel equipment [2]. Although its production increasingly improves our daily life, its negative impact on the environment cannot be ignored [3–5]. According to the World Health Organization [6], iron and steel manufacturing is a significant contributor to air pollution, especially in developing countries which have an increasing demand for steel in the domains of industrialisation and modernisation [7, 8].

China has witnessed a significant growth in steel production since 1996 [9]. Its total crude steel production was 928 million tonnes in 2018, contributing to 51.3% of the global crude steel production [10]. Airborne dust emissions in China were estimated at 8–17 million tonnes per year [11], with 27% resulting from iron and steelmaking industries [12]. The subsequent damage to the environment is reflected by the statistical analysis provided by China National Environmental Monitoring Centre, whereby the average $PM_{2.5}$ concentration produced by the top 20 cities in Chinese steel production list was 28% above the national average data [13]. For this reason, it is essential to better understand the airborne PM concentration and chemistry, and its potential health risks at iron and steelmaking areas in China in order to establish efficient control measures.

PM particles emitted from iron and steelmaking industries have been evidenced to include a range of toxic trace elements, posing a significant health risk [3, 14, 15]. Previous studies have found that the distribution of toxic trace elements is associated with the PM size [16–18]. For example, the trace elements As, Cd, Cr, Ni and Pb, which present significant health risks at low concentrations, tend to be accumulated in fine particles [19, 20]. However, the trace elements Fe and Zn, associated with negative health effects at high concentrations, were found accumulated in coarse PM particles [21]. Given the toxicity of trace elements varies, it is necessary to consider toxic factors for each individual element when investigating contamination levels at different particle sizes.

Nikolic [22] and Ilić [23] defined weight coefficients for key trace elements using the Preference Ranking Organization Method for Enrichment Evaluations (PROMETHEE) and Geometrical Analysis for Interactive Aid (GAIA) analysis to identify the contamination of smelting emissions in Europe. PROMETHEE and GAIA multivariate analysis methods have been widely used in the past to indicate the most preferred objects for decision making. For instance, they were used to rank contamination levels as a result of elevated trace elements at different sizes of sediment particles in Australia [24–27]. However, to date, PROMETHEE and GAIA multivariate analysis methods, which are based on toxic coefficients of trace elements, have not yet been used to estimate air quality across different atmospheric fractions.

The toxicity of an individual trace element is dependent on its state [28]. For example, airborne Cr has two primary forms of Cr(III) and Cr(VI) [29]. Trivalent Cr is considered to be an essential nutrient, while hexavalent Cr has been evidenced to be associated with an increased risk of lung and nasal cancer [30] and is classified by the U.S. Environmental Protection Agency [31] as 'Group A–Carcinogenic to Humans'. Cr and its compounds are widely used by the metallurgical industry, which is the dominant source of airborne Cr emissions into the environment [30]. However, previous studies used 1/6th of the total Cr concentrations as the hexavalent Cr for health risk assessment [32–36], and limited studies investigated the behaviour of extracted Cr(VI) concentrations in different atmospheric fractions coupled with assessment of their corresponding cancer risks at iron and steelmaking industrial areas [37].

PM particles are not only emitted from industrial sources but are also contributed by local traffic emissions or transported from remote sources as a result of atmospheric movement [6]. Pb isotopic composition analysis is an established tool to identify the contamination sources of trace elements in $PM_{10}$ and $PM_{2.5}$ emitted from coal burning, traffic emissions and metallurgical dust [38].

This study aims to investigate chemical composition and source apportionment of PM particles near iron and steelmaking industrial areas in China. The collected PM samples at different sizes were subjected to a range of chemical analyses, including trace element concentrations and bioavailability analyses as well as Cr(VI) extraction and Pb isotopic composition determination. The multicriteria analysis methods PROMETHEE and GAIA were applied to determine contamination levels across different atmospheric fractions. The potential health risks were assessed based on the extracted Cr(VI) concentrations coupled with other multiple trace elements near iron and steelmaking areas. The study further provides a Pb isotopic composition analysis to verify the influence of wind on contamination of atmospheric particles near ironmaking industrial sites. The results presented in this study are of significance to better understand and control PM emissions in these industrial areas.

## 2 Materials and methods

### 2.1 Sampling information

Four steel industrial areas investigated in this study were located at Kunming (KM) (24.8801˚ N, 102.8329˚ E), Wuhan (WH) (30.5928˚ N, 114.3055˚ E), Nanjing (NJ) (32.0603˚ N, 118.7969˚ E) and Ningbo (NB) (29.8683˚ N, 121.5440˚ E) (Fig 1). In addition, one background site at Ningbo Nottingham University (UN) was selected in this study (Fig 1). PM samples at five sampling sites were collected every 24 hours over a period of five days with similar environmental and meteorological conditions during April–July 2017. The temperature at KM, WH, NJ, NB and UN ranged from 10–26˚C, 18–33˚C, 19–34˚C, 17–28˚C and 18–28˚C, respectively, during the sampling period. The total precipitation was zero at KM, NJ and UN, while it was 5.6 mm at WH and 67 mm at NB which is a coastal city. The information of wind direction and speed at each sampling site is shown in Fig 1. The HYSPLIT backward trajectory at each sampling site was also provided to describe the origin of air masses and potential emission sources from remote areas during the sampling period (S1 Fig).

The steel mill at KM which was built in 1939 was the oldest industry among the four selected plants in this study, while the plant at NB was most recently established in 2003 and equipped with advanced particle capture instruments. The plants at WH and NJ were both build in 1958, but WH steel mill had the largest annual production of iron and steel at 20 million tonnes. Further information for the plant conditions of each industrial sampling location is detailed in Table 1.

### 2.2 Sample collection

Non-viable Andersen cascade impactor with 8 aluminium state plates (Model 20–800, Tisch Environmental) was installed at local meteorological sites or on the roof of high buildings to collect atmospheric particles at the five sampling sites shown in Fig 1. The 50% cut off diameters ($D_{50}$) of the Andersen sampler stages were 9.0, 5.8, 4.7, 3.3, 2.1, 1.1, 0.65 and <0.43 μm. Atmospheric particle samples (n = 200, eight samples per site per day) with different size fractions were collected from each sampling site for 24 h loading at a flow of 28.3 L/min. All the samples were stored at 4˚C until analysis.

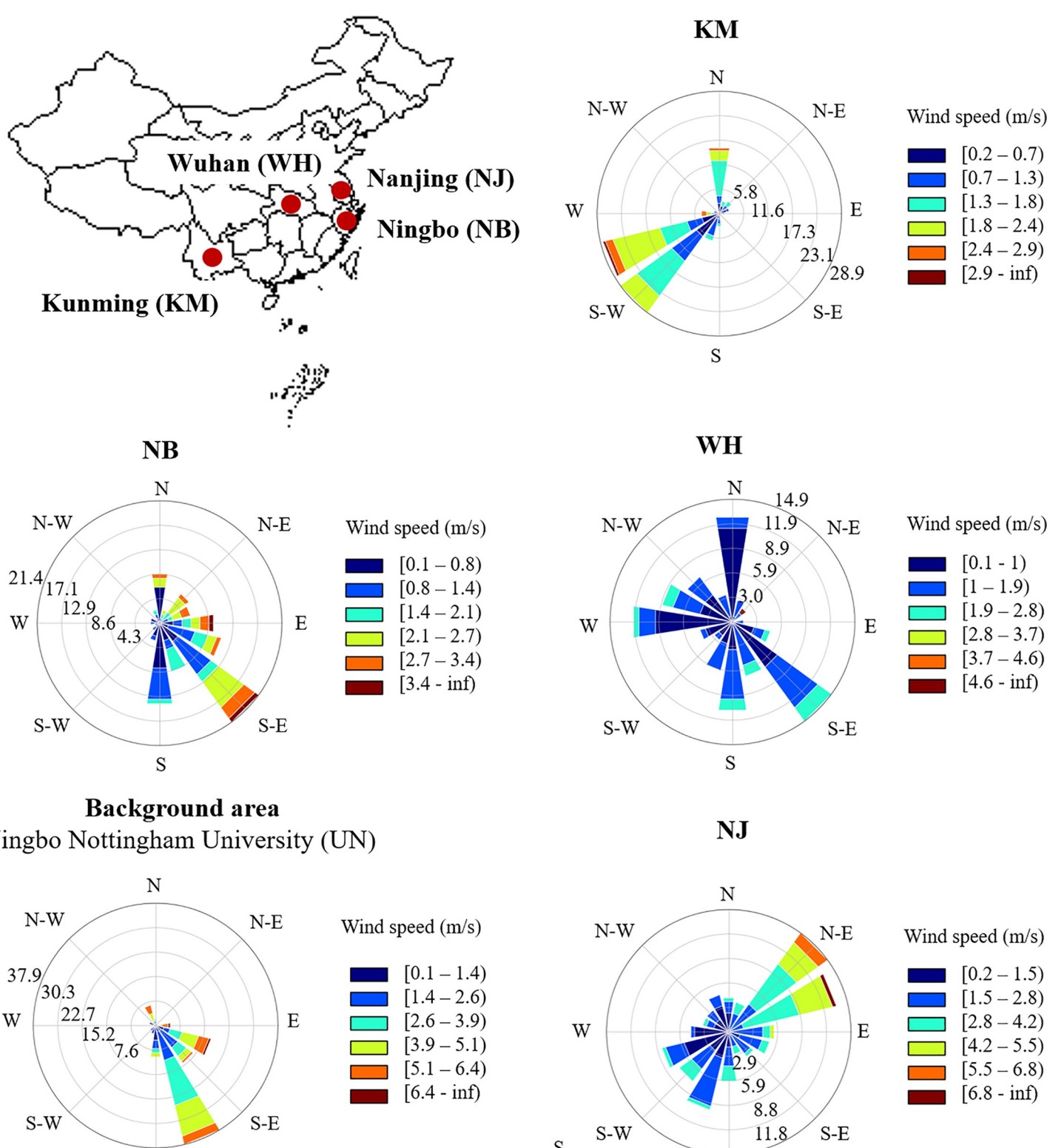

**Fig 1. Sampling locations and wind information at four steel industrial areas in Kunming (KM), Wuhan (WH), Nanjing (NJ) and Ningbo (NB) and the background area located at Ningbo Nottingham University (UN).** The industrial plant at each sampling site and the UN campus were outlined with green lines and red star symbols to highlight the sampling locations. The sampling locations at four industrial areas were within < 1 km away from the steel plants.

**Table 1. Industrial information on iron and steel plants at Kunming (KM), Wuhan (WH), Nanjing (NJ) and Ningbo (NB) in China.**

| | | KM | WH | NJ | NB |
|---|---|---|---|---|---|
| Plants | Established year | 1939 | 1958 | 1958 | 2003 |
| | Iron and steel production per year (million tons) | 7 | 20 | 9 | 4 |
| | No. of blast furnace (BF)* | 1 | 8 | 5 | 2 |
| | No. of blast oxygen furnace (BOF)* | 3 | 10 | 6 | 3 |
| | No. of electric arc furnace (EAF)* | 0 | 0 | 1 | 0 |
| | Air pollution control measurements | 1) Limestone gypsum flue gas desulfurization; 2) Electrostatic precipitators; 3) Bag filters; | 1) Limestone gypsum flue gas desulfurization; 2) Electrostatic precipitators; 3) Bag filters; 4) Wet scrubber; | 1) Limestone gypsum flue gas desulfurization; 2) Bag filters; 3) Wet scrubbers; 4) Long bag type pulse filters; | 1) Electrostatic precipitators; 2) Bag filters; 3) Wet scrubbers; |
| | Recirculation system | No | No | No | Flue gas recirculation |
| Sampling locations of plants | | Downwind | Downwind | Downwind | Upwind |

*Facility information was obtained from Yang [39].

## 2.3 Laboratory analysis

A five 24-hour sampling protocol was conducted for each sampling sites. Filters from two days of sampling were subjected for Cr(VI) analysis, filters from the third day were used for bio-availability analysis and the last two-day samples were used for multiple trace element analysis and Pb isotopic composition analysis. All the samples were analysed using an inductively coupled plasma optical emission spectrometer (ICP-OES, Varian 730-ES) and an inductively coupled plasma mass spectrometer (ICP-MS, Agilent 7900) at the Australian National Measurement Institute, Sydney, Australia.

**2.3.1 Cr(VI) analysis.** Air filter samples with $PM_{2.1-9.0}$, $PM_{1.1-2.1}$, and $PM_{1.1}$ size fractions across five sampling sites over two days were selected for Cr(VI) analysis. Samples were digested using a $Na_2CO_3$/NaOH solution and heated at 100°C for three hours to extract Cr (VI) and stabilize against reduction to Cr(III). The solution was diluted 10 times using 1 mL of supernatant adding 0.3 mL $HNO_3$ (15.6 M) and making up to 10 mL with Milli Q water (18.2 MΩ·cm) prior to ICP-MS analysis. A water insoluble Cr(VI) blank spike (0.201 mg/L $PbCrO_4$) and a Cr(III) blank spike (0.98 mg/L $CrCl_3 \cdot 6H_2O$) were used to assess the accuracy of the extraction procedure for Cr(VI). The recovery rates of spike samples ranged from 91 to 109%.

**2.3.2 Bioavailability analysis.** In order to assess ingestion health risks caused by trace element concentrations at different atmospheric particle sizes, the bioavailability analysis for trace elements was carried out in this study. All the air filters were folded and placed into 12 mL of graduated polypropylene centrifuge tubes. 10 mL of HCl (1 M) was added before tumbling for 1 hour. All the samples were diluted twice prior to ICP-MS analysis. Trace element concentrations for Al, As, Ba, Bi, Cd, Cr, Cu, Fe, La, Mn, Pb, Sb, Sr, V and Zn were determined for their bioavailable concentrations via ingestion exposure. Their procedural blanks were below the Limit of Reporting (LOR) of < 0.01 mg/kg and the recovery rates ranged from 96 to 116%.

**2.3.3 Multiple trace element analysis.** The filter samples were digested using HCl and $HNO_3$ (1:3, respectively, 8 ml) for 2 h. Each digested sample was topped up to 40 mL with Milli-Q water. Samples were diluted twice prior to analysis for Al, Ca, K, Mg and Na concentrations on an ICP-OES instrument. Concentrations of As, Ba, Bi, Cd, Ce, Cr, Cu, Fe, La, Mn, Pb, Sb, Sr, Ti, V, W, Zn and Zr were analysed using an ICP-MS instrument. Each sample batch

(n = 20) contained a filter blank and duplicate, blank spike, blank matrix and matrix spikes. Procedural blanks were below the LOR of < 0.05 mg/kg for Al, Ca, K, Mg and Na, and < 0.01 mg/kg for As, Ba, Bi, Cd, Ce, Cr, Cu, Fe, La, Mn, Pb, Sb, Sr, Ti, V, W, Zn and Zr. Recovery rates for Al, Ca, K, Mg and Na ranged between 93–105% for all samples. Recovery rates for As, Ba, Bi, Cd, Ce, Cr, Cu, Fe, La, Mn, Pb, Sb, Sr, Ti, V, W, Zn and Zr for air filters were 95–110%.

**2.3.4 Pb isotopic composition analysis.** Atmospheric samples collected from KM ($PM_{2.1-9.0}$, $PM_{1.1-2.1}$, $PM_{1.1}$), WH ($PM_{2.1-9.0}$, $PM_{1.1-2.1}$, $PM_{1.1}$), NJ ($PM_{2.1-9.0}$, $PM_{1.1-2.1}$, $PM_{1.1}$), NB ($PM_{1.1}$) and UN ($PM_{1.1}$) were subjected to Pb isotopic composition analysis ($^{204}Pb/^{207}Pb$, $^{206}Pb/^{207}Pb$, $^{208}Pb/^{207}Pb$) after sample volumes were optimized on the basis of their Pb concentrations. The National Institute of Standards and Technology SRM981 was used to correct mass fractionations, and certified values of SRM981 for $^{204}Pb/^{207}Pb$, $^{206}Pb/^{207}Pb$ and $^{208}Pb/^{207}Pb$ are $0.0646 \pm 0.000047$, $1.0933 \pm 0.00039$, and $2.3704 \pm 0.0012$, respectively. The mean RSDs for sample analysis $^{204}Pb/^{207}Pb$, $^{206}Pb/^{207}Pb$ and $^{208}Pb/^{207}Pb$ were 0.43%, 0.26% and 0.16%, respectively.

## 2.4 Data analysis

**2.4.1 PROMETHEE and GAIA.** PROMETHEE is a non-parametric method capable of ranking objects from the most preferred to the least preferred. In this study, PROMETHEE was applied to indicate the most polluted atmospheric particles and the most polluted sampling sites based on multiple variables.

The V shape preference function was selected for all the trace elements based on the maximum values as preference thresholds for each variable. The weights for the trace elements were referred to the toxicity points of each trace element outlined in Substance Priority List of Agency for Toxic Substances and Disease Registry (ATSDR 2017). The details of modelling parameters, including weight coefficients, preference functions, thresholds and trace element concentrations, were compiled in the S1 Table for the PM fractions at 2.1–9.0 μm, 1.1–2.1 μm and < 1.1 μm.

GAIA is a descriptive complement to the PROMETHEE rankings. It provides more profound insight into the relations between samples ranked with PROMETHEE and information on the variables responsible for the ranking. In a GAIA biplot, variables are considered to have positive correlations if their projected vectors form acute angles, negative correlations if they form obtuse angles, and no correlation if they are orthogonal [40, 41].

**2.4.2 Health risks.** The exposure concentration ($EC_{inhalation}$) and average daily dose ($ADD_{ingestion}$) at three fractions of $PM_{2.1-9.0}$, $PM_{1.1-2.1}$ and $PM_{1.1}$ were calculated according to Eqs 1 and 2.

$$EC_{inhalation} = C_{inhalation} \times \frac{ET \times EF \times ED}{AT \times 24} \times CF \qquad \text{Eq1}$$

$$ADD_{ingestion} = C_{ingestion} \times \frac{IngR \times EF \times ED}{BW \times AT} \times CF \qquad \text{Eq2}$$

Where $C_{inhalation}$ refers to the total concentrations of trace elements for the PM fractions $PM_{2.1-9.0}$, $PM_{1.1-2.1}$ and $PM_{1.1}$ (ng/m³, S1 Table); $C_{ingestion}$ refers to trace element concentrations extracted from the bioavailability analysis (mg/kg, S2 Table); ET is exposure time (ET = 24 hours/day in this study); EF is exposure frequency (EF = 365 days/year in this study); ED is exposure duration (ED = 24 years in this study); AT is the average time (AT = ED × 365 days for non-carcinogens, AT = 70 × 365 days for carcinogens); IngR is the ingestion rate of 100 mg/day for adults [42]; BW is the average body weight (BW = 66.1 kg for male and BW = 57.8 for female) [43]; CF is the conversion factor of $10^{-6}$.

The health risks were characterised by non-carcinogenic and carcinogenic effects using equations of hazard quotient (HQ) and cancer risk (CR), respectively (Eqs 3 and 4).

$$HQ = EC/RfCi = ADD_{ingestion}/RfDo \qquad \text{Eq3}$$

$$CR = EC \times IUR = ADD_{ingestion} \times SFo \qquad \text{Eq4}$$

RfCi is chronic inhalation reference concentration (mg/m$^3$); RfDo is chronic oral reference dose (mg/kg·day); IUR is inhalation unit risk (μg/m$^3$)$^{-1}$; SFo is oral slope factor (mg/kg·day)$^{-1}$. All the parameters used in Eqs 3 and 4 were obtained from U.S. EPA [44].

When the values of HQ for individual trace elements or the sum of HQ (ΣHQ) are higher than one, then a chance exists for non-carcinogenic effects due to inhalation or ingestion exposure [45]. The tolerable value of CR at $10^{-6}$ means that the risk of developing cancer over a human lifetime (70 years) is one out of 1,000,000 people.

## 3 Results

### 3.1 PROMETHEE and GAIA

The PROMETHEE analysis was used to determine the most detrimental particle fraction and the site at which this occurred. According to Table 2, the PM$_{1.1}$ at three of the industrial sampling sites at NJ, WH and NB and the background site UN had higher Phi values than the larger two fractions of PM$_{1.1-2.1}$ and PM$_{2.1-9.0}$ for the corresponding locations, suggesting that the fine atmospheric particles were the most polluted fractions, regardless of the industrial activity.

The GAIA analysis was used to reveal correlations between the trace elements and the associate sampling site. A decision axis (Pi) displayed in a GAIA plane was indicative of the most polluted sampling site. The length and quality numbers were used to evaluate the reliability of the result represented in a GAIA plane.

Criteria vectors of the trace elements at the coarse fractions of PM$_{2.1-9.0}$ were oriented in the same direction with the decision axis of Pi pointing towards the sampling site of KM (Fig 2). This suggests that KM had the most polluted PM$_{2.1-9.0}$ particles than the other sampling

**Table 2. PROMETHEE ranking of PM$_{2.1-9.0}$, PM$_{1.1-2.1}$, and PM$_{1.1}$ at sampling sites KM, WH, NJ, NB and UN with Phi, Phi+ and Phi- values.**

| Rank | Samples | Locations | Phi | Phi+ | Phi- |
|---|---|---|---|---|---|
| 1 | PM$_{1.1}$ | NJ | 0.46 | 0.50 | 0.04 |
| 2 | PM$_{1.1}$ | WH | 0.38 | 0.42 | 0.04 |
| 3 | PM$_{2.1-9.0}$ | KM | 0.34 | 0.39 | 0.05 |
| 4 | PM$_{1.1}$ | KM | 0.24 | 0.30 | 0.06 |
| 5 | PM$_{2.1-9.0}$ | WH | 0.04 | 0.15 | 0.10 |
| 6 | PM$_{1.1}$ | UN | -0.03 | 0.09 | 0.12 |
| 7 | PM$_{1.1}$ | NB | -0.05 | 0.07 | 0.13 |
| 8 | PM$_{1.1-2.1}$ | KM | -0.12 | 0.05 | 0.17 |
| 9 | PM$_{2.1-9.0}$ | NB | -0.13 | 0.04 | 0.17 |
| 10 | PM$_{2.1-9.0}$ | NJ | -0.14 | 0.04 | 0.18 |
| 11 | PM$_{2.1-9.0}$ | UN | -0.15 | 0.03 | 0.18 |
| 12 | PM$_{1.1-2.1}$ | WH | -0.15 | 0.03 | 0.18 |
| 13 | PM$_{1.1-2.1}$ | NJ | -0.18 | 0.02 | 0.20 |
| 14 | PM$_{1.1-2.1}$ | NB | -0.23 | 0.00 | 0.24 |
| 15 | PM$_{1.1-2.1}$ | UN | -0.26 | 0.00 | 0.26 |

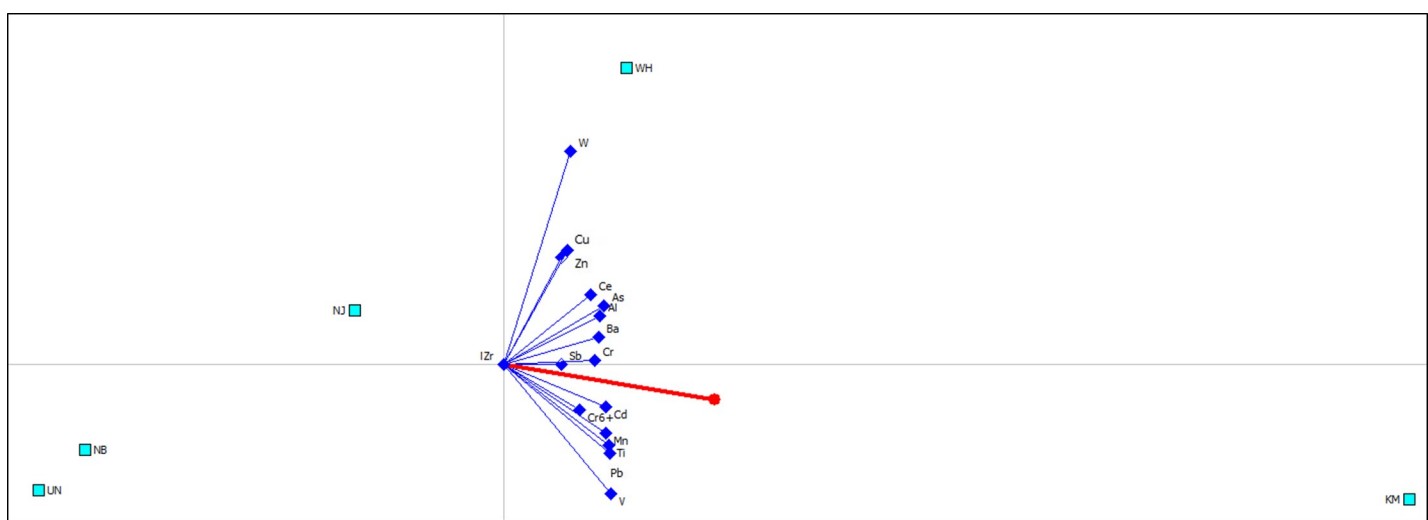

**Fig 2. GAIA biplot for trace element contamination at atmospheric fraction of PM$_{2.1-9.0}$ with quality of 94.5%.** The decision axis (Pi) is displayed with a red line, and the ranking of Phil values obtained from PROMETHEE analysis at fives sampling sites is KM (0.7227) > WH (0.0805) > NJ (-0.1785) > NB (-0.2974) > UN (-0.3273).

sites, which was consistent with the PROMETHEE results which demonstrated that PM$_{2.1-9.0}$ at KM had Phi value of 0.72 while the other samples had Phi values ranging from -0.33 to 0.08 (Fig 2). The industrial site NB and the background site UN were clearly grouped together and were in an obvious conflict with all criteria, indicating that both sites had the least polluted coarse particles.

Similarly, the sites NB and UN were also found to be located in the opposite direction to the Pi axis in Fig 3, suggesting that the fine PM$_{1.1-2.1}$ particles at both sites were the least polluted amongst all the sampling sites. PM$_{1.1-2.1}$ fractions for WH and KM had Phi values of 0.42 and 0.31, respectively, indicating pollution where the PM$_{1.1-2.1}$ at WH were highly correlated with trace elements Cr, Cd, Ce, W, As and Cu, while at KM they were associated with Pb, V, Ti, Mn and Al (Fig 3).

Compared to GAIA biplots of PM$_{2.1-9.0}$ and PM$_{1.1-2.1}$ (Figs 2 and 3), the GAIA results of PM$_{1.1}$ had a more heterogeneous distribution of trace elements (Fig 4). The most polluted fraction of PM$_{1.1}$ was found for the NJ site, which had the highest Phi value of 0.30 (Fig 4). The sampling site KM was independent from other sites and was associated with the trace element Ti (Fig 4). Trace elements As, Cd and Pb with long axes were close to the decision maker Pi, suggesting these elements played a significant role in the contamination levels of the fine particles.

### 3.2 Pb isotopic compositions

The Pb compositions of $^{206}$Pb/$^{207}$Pb and $^{208}$Pb/$^{207}$Pb in the PM samples at all sampling sites are plotted in Fig 5. The PM data in this study was compared to other potential end-members, including uncontaminated background Pb [46], coal combustion [47], metallurgical dust [47], as well as traffic emissions associated with leaded petrol [47], unleaded petrol [47] and other automobile exhaust [46].

The Pb isotopic compositions of atmospheric particles collected near the iron and steelmaking areas of KM, WH, NJ and NB were distinct from the background Pb and traffic Pb, but very close to the Pb signatures of coal combustion and metallurgical dust (Fig 5).

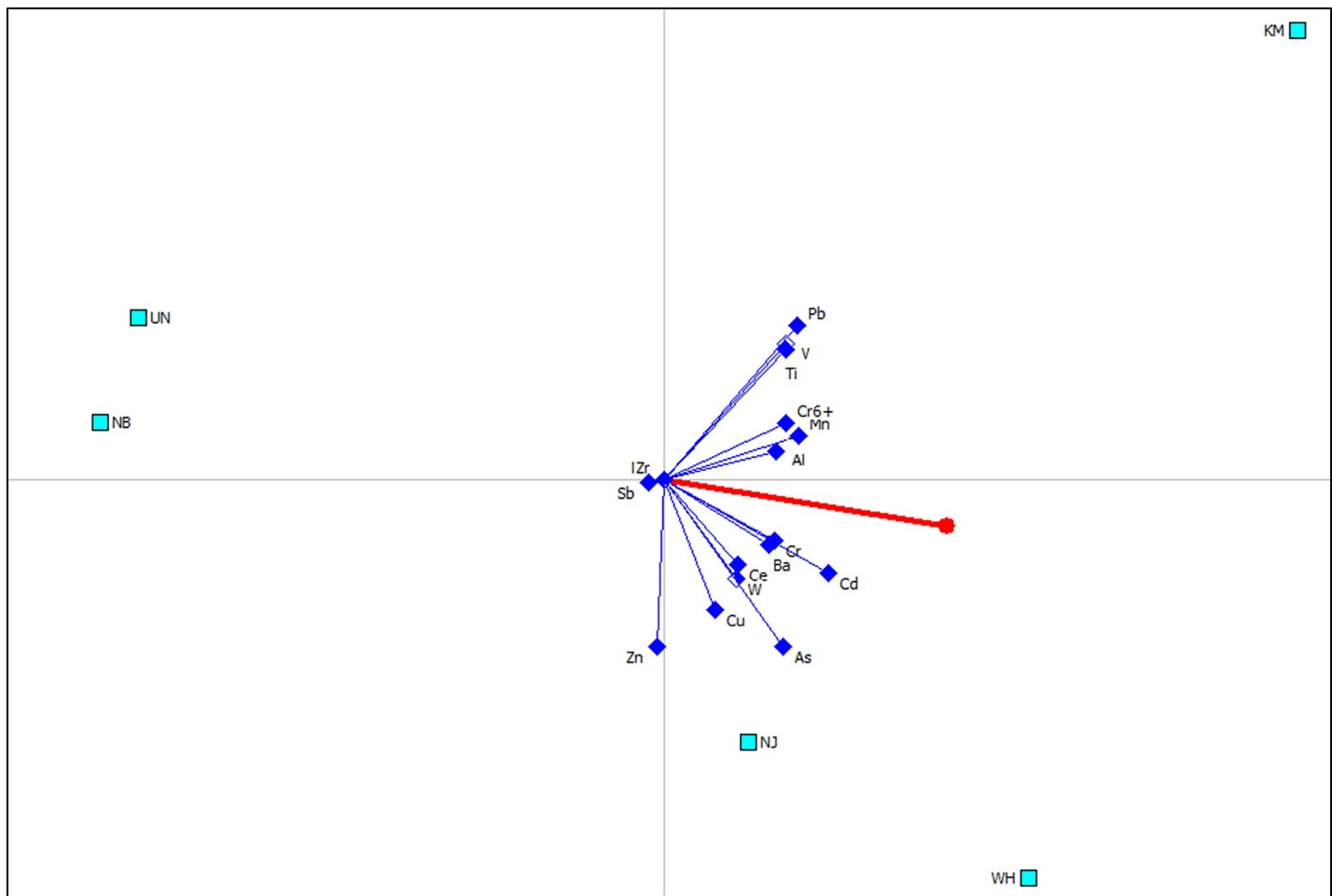

**Fig 3. GAIA biplot for trace element contamination at atmospheric fraction of PM$_{1.1-2.1}$ with quality of 82.5%.** The decision axis (Pi) is displayed with a red line, and the ranking of Phil values obtained from PROMETHEE analysis at fives sampling sites is WH (0.4185) > KM (0.3059) > NJ (0.0932) > NB (-0.333) > UN (-0.4843).

In order to determine Pb sources across different particle fractions, the Pb isotopic compositions of $^{206}$Pb/$^{207}$Pb and $^{208}$Pb/$^{207}$Pb at PM$_{2.1-9.0}$, PM$_{1.1-2.1}$ and PM$_{1.1}$ collected from KM, WH and NJ were plotted in Fig 6. The Pb isotopic compositions of PM$_{2.1-9.0}$, PM$_{1.1-2.1}$ and PM$_{1.1}$ at KM site were overlapped, suggesting that the Pb contamination across three fractions at KM were dominated by the same source, such as the local steelmaking activities. However, the Pb isotopic compositions at PM$_{2.1-9.0}$ and PM$_{1.1}$ fractions showed distinct differences at the sampling sites of WH and NJ, indicating that the coarse and fine particles have different dominant Pb contamination sources.

## 3.3 Health risks

**3.3.1 Inhalation risks.**   According to Eqs 3 and 4, the reference values of RfCi were used to calculate non-carcinogenic risks (HQ) for each trace element. The RfCi for trace elements Al, As (inorganic), Ba, Cd, Cr(VI), Mn and V were 5.0, 0.015, 0.5, 0.01, 0.1, 0.05 and 0.1 μg/m$^3$, respectively [44], and their corresponding non-carcinogenic risks (HQ and ΣHQ) were calculated for three atmospheric fractions (PM$_{2.1-9.0}$, PM$_{1.1-2.1}$ and PM$_{1.1}$) at five sampling sites (Fig 7A).

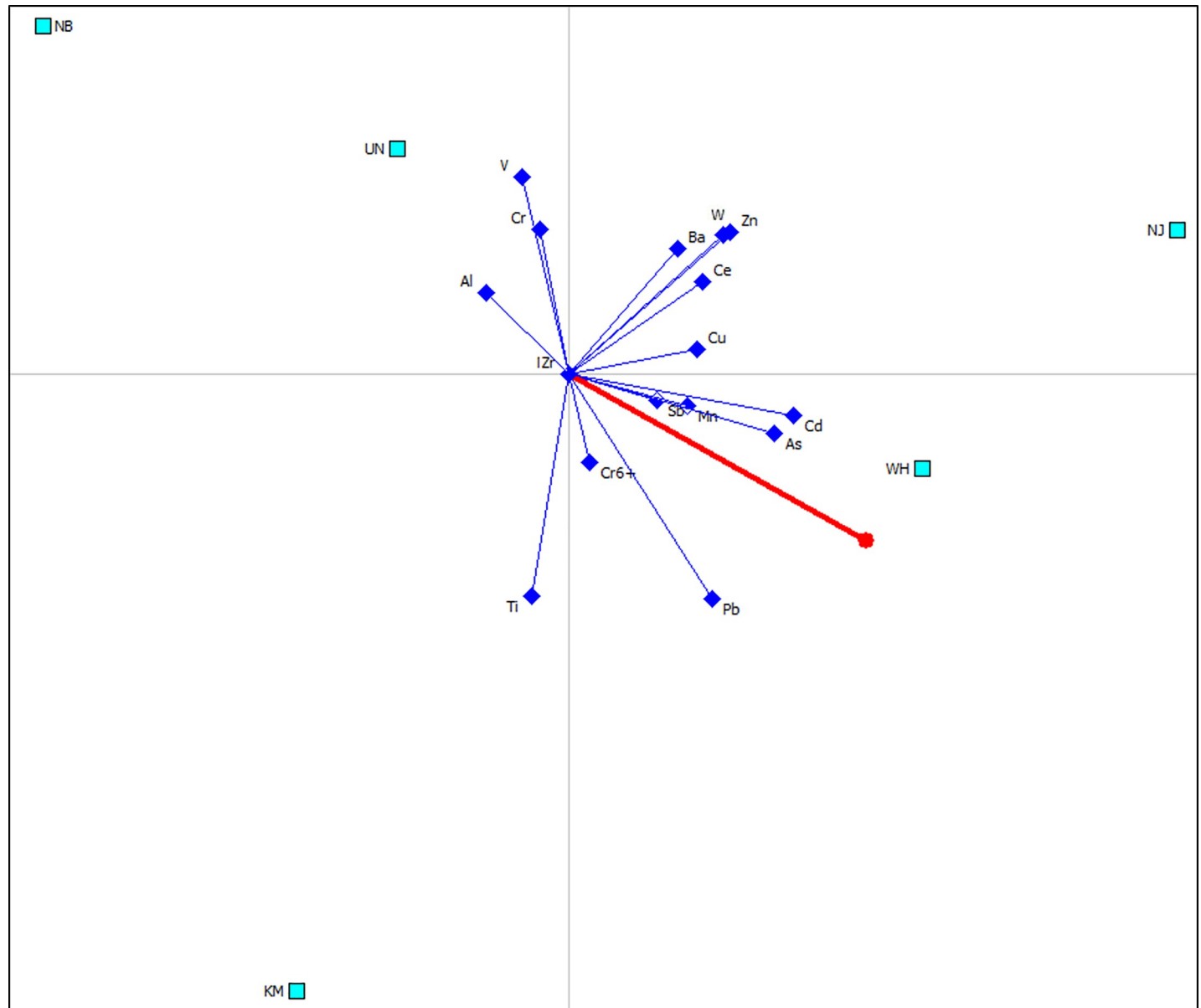

**Fig 4. GAIA biplot for trace element contamination at atmospheric fraction of PM$_{1.1}$ with quality of 73.2%.** The decision axis (Pi) is displayed with a red line, and the ranking of Phil values obtained from PROMETHEE analysis at fives sampling sites is NJ (0.3045) > WH (0.2480) > KM (0.0421) > UN (-0.2911) > NB (-0.3035).

The highest $\Sigma$HQ value of 5.32 was found for the fraction of PM$_{2.1-9.0}$ collected from KM, with As and Mn as the dominant contributing trace elements which had HQ values of 1.88 and 2.77, respectively (Fig 7A). These values were much higher than the safe level of HQ = 1, suggesting that concentrations of As and Mn coupled with other trace elements at PM$_{2.1-9.0}$ at KM posed a significant non-carcinogenic risk.

The coarse fraction of PM$_{2.1-9.0}$ at WH ($\Sigma$HQ = 1.06) and fine fraction of PM$_{1.1}$ at NJ ($\Sigma$HQ = 1.08) also had $\Sigma$HQ values beyond the limit of one, but there was no individual trace element with HQ > 1. This suggests that trace element concentrations of PM$_{2.1-9.0}$ at WH and PM$_{1.1}$ at NJ can result in the accumulative non-carcinogenic health risk for residents via inhalation exposure.

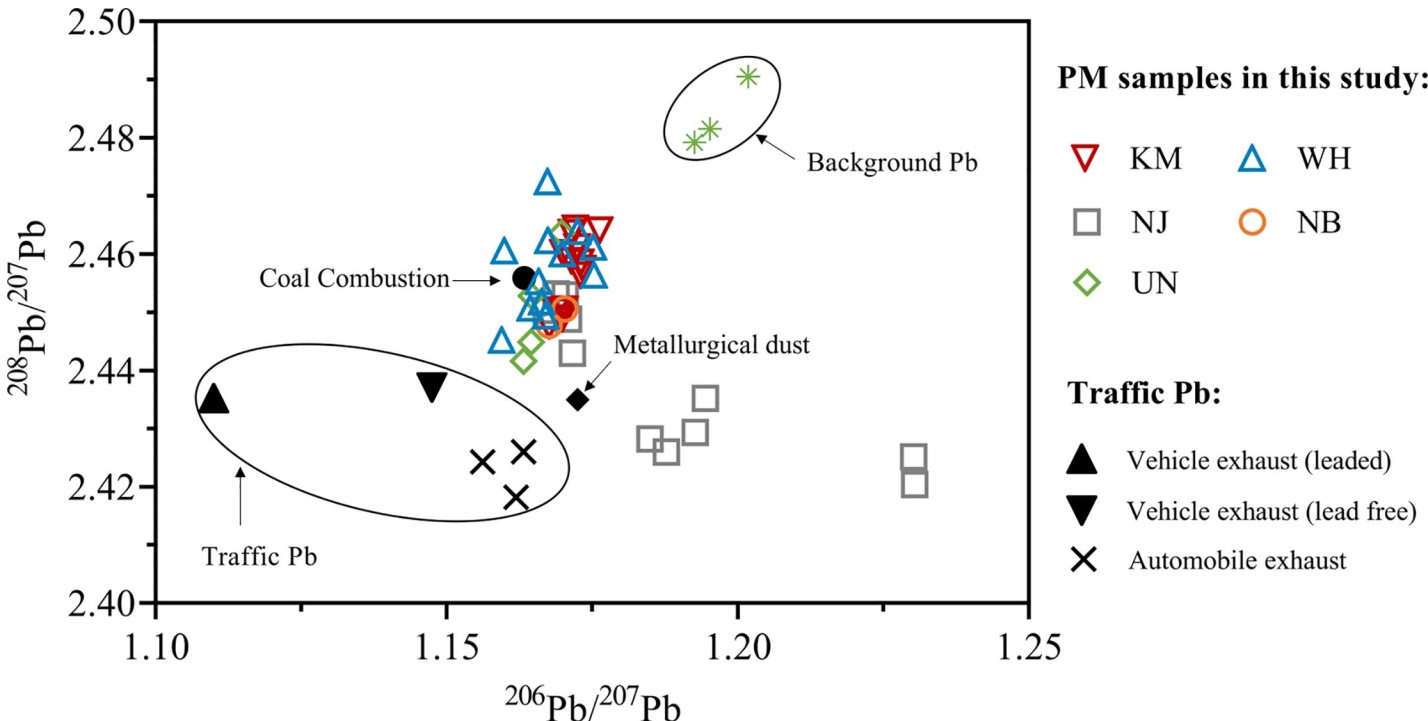

**Fig 5. Lead isotopic compositions (²⁰⁶Pb/²⁰⁷Pb, ²⁰⁸Pb/²⁰⁷Pb) for PM samples collected at KM, WH, NJ, NB and UN.** Lead isotopic compositions for background, coal combustion, metallurgical dust and traffic sources were obtained from Zhou et al. (2001) [46], and Tan [47].

Cancer risks via inhalation exposure were calculated and displayed in Fig 7B. The IUR values for trace elements As (inorganic), Cd and Cr(VI) are 4.3E-03, 1.8E-03 and 8.4E-02 (µg/m³)⁻¹, respectively [44], and hence these three elements were used to assess the potential carcinogenic risks. In order to determine a relative conservative health risk caused by As, the total concentrations of As measured in this study were assumed to be in the inorganic form [36], which is considered to be associated with cardiovascular disease and diabetes [48].

The CR value between $10^{-6}$ and $10^{-4}$ is considered to pose a potential cancer risk and CR values with $> 10^{-4}$ are considered to highly likely cause cancer. In this study, apart from the Cr

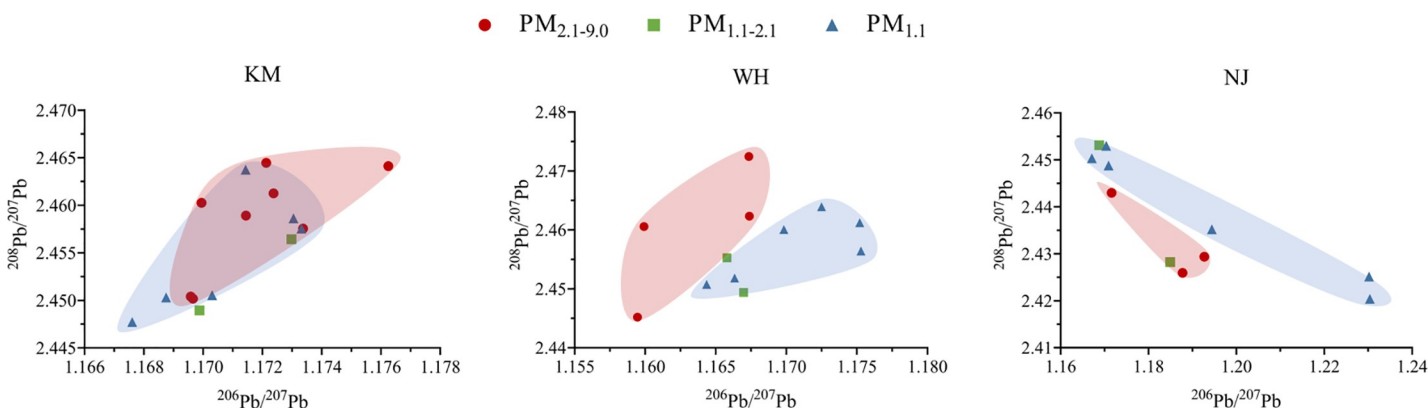

**Fig 6. Lead isotopic compositions (²⁰⁶Pb/²⁰⁷Pb, ²⁰⁸Pb/²⁰⁷Pb) for different atmospheric fractions of PM₂.₁₋₉.₀, PM₁.₁₋₂.₁ and PM₁.₁ at sampling sites of KM, WH and NJ.**

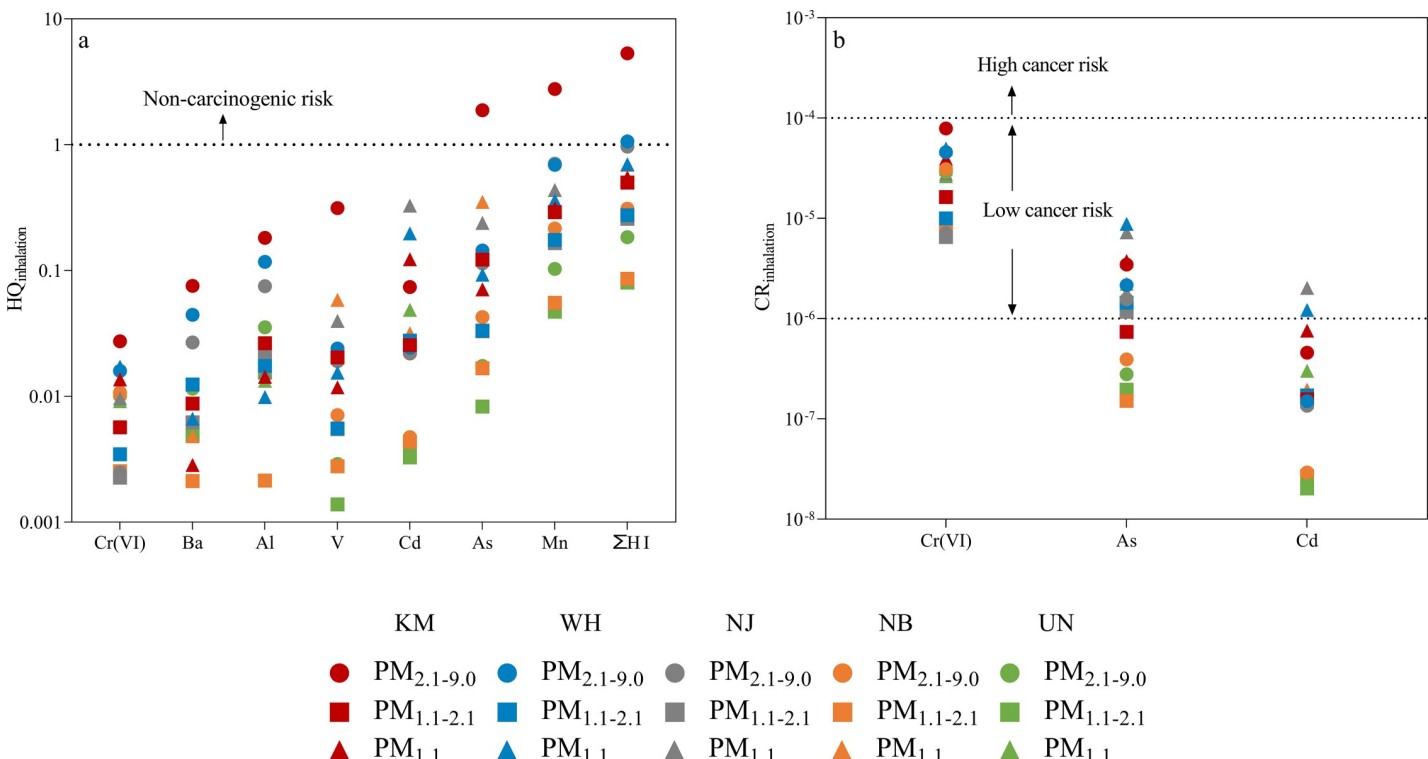

**Fig 7. Non-carcinogenic risk and carcinogenic risk via inhalation exposure at fractions of $PM_{2.1-9.0}$, $PM_{1.1-2.1}$ and $PM_{1.1}$ across five sampling sites.** The absence of symbols for HQ and CR values was due to the corresponding trace elements with concentrations < LOR.

(VI) of $PM_{1.1-2.1}$ fraction at NB site with a concentration < LOR and not used for cancer risk assessment, Cr(VI) concentrations of the other samples presented high CR values exceeding the limit of $10^{-6}$ (Fig 7B). This suggests that airborne Cr(VI) near the iron and steelmaking plants investigated in this work pose a likely cancer risk to humans.

In addition to Cr(VI), As was also found to have CR values greater than $10^{-6}$ at different PM fractions, especially in the fine particle of $PM_{1.1}$. The $PM_{1.1}$ samples at all the sampling sites were found to have carcinogenic risks as a result of As contamination. The health risks caused by Cd contamination were also found at $PM_{1.1}$ particles (Fig 7B), indicating that the fine particles posed more adverse inhalation risks than coarse particles.

**3.3.2 Ingestion risks.** Trace element concentrations extracted from the bioavailability analysis were used to assess the non-carcinogenic and carcinogenic health risks via ingestion exposure. The parameter of body weight (BW) used to calculate the health risks were different for males and females, hence the average daily dose (ADD), hazard quotient (HQ) and cancer risk (CR) values were displayed according to genders (Fig 8A–8C). The results showed that Mn, Pb, Zn, Al and Fe were the dominant trace elements for ADD ingestion values, and that females tend to have a higher exposure dose via ingestion than males (Fig 8A). As a result, females experienced slightly higher non-carcinogenic risks than males (Fig 8B).

The HQ results suggested that the fine particles $PM_{1.1}$ had the highest HQ values, although their HQ values and the sum of HQ values did not exceed the safety limit of HQ = 1. A similar trend was also found for carcinogenic risk assessment. The $PM_{1.1}$ particles had higher CR values than $PM_{1.1-2.1}$ and $PM_{2.1-9.0}$ particles for both genders, although their values were within the acceptable limit of $10^{-6}$ (Fig 8C).

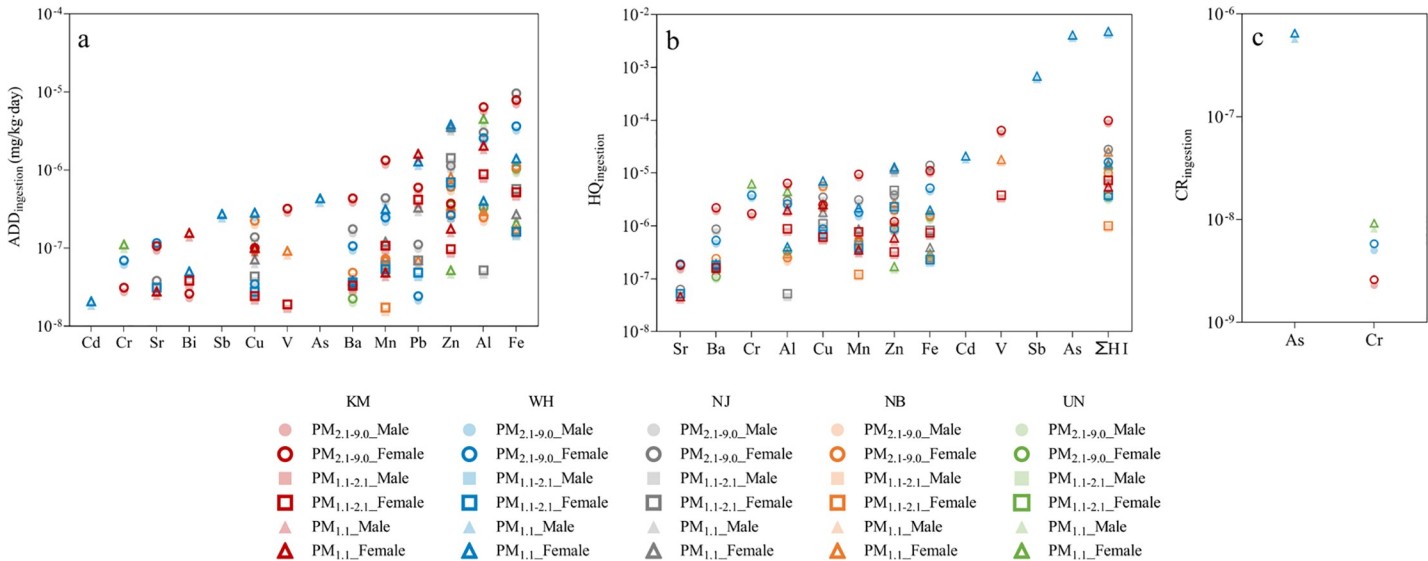

**Fig 8. Data for average daily dose (ADD), hazard quotient (HQ) and cancer risk (CR) via ingestion exposure for $PM_{2.1-9.0}$, $PM_{1.1-2.1}$ and $PM_{1.1}$ across five sampling sites.** The absence of symbols for ADD, HQ and CR values was due to the corresponding trace elements with concentrations < LOR. The health risks caused by Cr contamination were calculated by $1/6^{th}$ of the total bioavailable Cr concentrations, while the total bioavailable As concentrations were assumed as an inorganic form [36].

## Discussion

The iron and steelmaking activities are considered as a significant contributor to the state of local air quality in China [13]. This study showed that trace element concentrations at PM particles collected from intensive steelmaking areas in China were elevated with higher risks to human health than samples at the background site located away from industrial activities (S1 Table, Figs 7 and 8). For example, the trace elements As and Mn in $PM_{2.1-9.0}$ fractions at KM site were 12 and 27 times higher than the corresponding concentrations of the background UN sample. As a result, the elevated As and Mn contents at $PM_{2.1-9.0}$ of KM were found to exceed the safety limits, presenting non-carcinogenic risks via inhalation exposure (Fig 7A) [49].

Compared to the air quality at steelmaking areas of KM, WH and NJ, PM particles collected near the steel mill at NB had low trace element concentrations, which were similar to the levels at the background area of UN (S1 Table). The PROMETHEE and GAIA results clearly showed that the sampling sites of NB and the background UN can be grouped into a same cluster for the three fractions of $PM_{2.1-9.0}$ (Fig 2), $PM_{1.1-2.1}$ (Fig 3) and $PM_{1.1}$ (Fig 4) because the sampling site at NB was located at the upwind of the local steel mill (Table 1), and was approximately 40 km away from the UN site.

The trace element analysis also showed that the small PM particles tend to be more contaminated than the coarse fractions [50]. This was also found in the PROMETHEE results which showed that the fine particles at $PM_{1.1}$ size had higher Phi values than the other two fractions (Table 2), suggesting higher enrichment by trace elements. As a result, fine particles of $PM_{1.1}$ presented a higher health risk than $PM_{1.1-2.1}$ and $PM_{2.1-9.0}$ [33]. For example, concentrations of As, Cd and Cr(VI) at $PM_{1.1}$ fractions across the steelmaking areas had CR values up to 4.96E-05, indicating that approximately five people out of a population of 100,000 have possibility to develop cancer during the lifetime of 70 years.

The PM particles incorporated with trace elements are generated during various stages of the steelmaking process [51]. The BOF and BF as key facilities for steelmaking are considered

as a large contributor to PM emissions [52]. The steel mill at WH investigated in this study has the maximum number of furnaces (8 BFs and 10 BOFs) with the largest capacity for steel production of up to 20 million tonnes per year among the sampled plants (Table 1). Consequently, the sampling site of WH presented acute angles with the decision axis Pi in GAIA biplots, suggesting PM particles at WH were highly contaminated by trace elements across the three fractions, especially $PM_{1.1-2.1}$ and $PM_{1.1}$ (Figs 3 and 4). GAIA results further showed that PM emission at WH were significantly associated with As and Cd at fine particles as both elements presented long criterial axes pointing towards WH (Figs 3 and 4). Previous studies have evidenced that As and Cd emissions at fine PM particles were related to metal smelting and fuel combustion under high temperature processing [39]. Previous studies also showed that chemical compositions of airborne particle emissions varied with steelmaking facilities [53]. For example, compared to BOF dust, which was estimated to contain 2–8% Zn [54], the EAF dust contains Zn up to 19.4% [55]. This high correlation between Zn contamination and EAF facility was also observed in this study. The only EAF equipment in this study was located at the NJ steel mill (Table 1), and the vector of Zn concentrations presented a long criterial axis highly related to its local $PM_{1.1-2.1}$ and $PM_{1.1}$ fractions (Figs 3 and 4).

Although iron and steelmaking activities were a dominant source to PM emissions, traffic emission also played a significant role in local air quality [56]. Traffic emissions related to brake wear, road abrasion and dust resuspension as a result of the mechanical process are considered a significant contributor to PM coarse fractions [57], which is also relevant to this study. At the KM site, the sampling location was next to a main road with high traffic volume, while sampling locations at WH, NJ and NB sites were far away from other contamination sources. As a result, the coarse fractions of $PM_{2.1-9.0}$ at KM had higher trace element concentrations with higher Phi value (Phi = 0.34) than $PM_{2.1-9.0}$ particles at other sampling sites (Phi = -0.15–0.04) (Table 2). The GAIA results also showed that the sampling site of KM was consistent with the decision axis Pi and far away from other sites at $PM_{2.1-9.0}$ fraction (Fig 2). This suggests that the coarse fraction at KM was most contaminated among the sampling sites as a result of local contamination sources of both iron and steelmaking activities and heavy traffic.

In addition to the local contamination sources, PM particles can also be transferred from the remote pollution sources (S1 Fig), particularly the fine $PM_{1.1}$ particles which can travel up to tens of kilometres [58]. The remote contributor to $PM_{1.1}$ contamination was also observed in this study. Compared to coarse fraction of $PM_{2.1-9.0}$ and intermodal $PM_{1.1-2.1}$, the small particles of $PM_{1.1}$ had a more heterogenous distribution of trace elements in the GAIA plot (Fig 4 vs Figs 2 and 3). This indicated relatively low interrelationships between trace elements at $PM_{1.1}$ fractions due to various remote sources in addition to local iron and steelmaking contamination.

The main driving factor bringing remote pollution into the local environment is hypothesised to be wind (S1 Fig). Compared to other sampling sites with clear weather during the sampling period, the NJ site experienced strong winds with the maximum speed of 6.8 m/s and frequencies ranging from 2.8 to 5.5 m/s (Fig 1). As a result, the particles in this site had larger $^{206}Pb/^{207}Pb$ and $^{208}Pb/^{207}Pb$ range at $PM_{1.1}$ than in $PM_{2.1-9.0}$, suggesting that Pb contamination in the fine $PM_{1.1}$ particles at NJ was not only dominated by local iron and steelmaking activities but contributed by remote contamination through the forces of strong winds.

## Conclusion

This study collected PM particles across different sizes near iron and steelmaking areas in China and characterized the PM fractions with multiple lines of chemical and statistical analyses. Due to the complexity of performing atmospheric sampling in industrial areas in China,

the objectives of the study were limited to assessment of the application of the selected analytical and modelling research methods to study differences of atmospheric particles in industrial areas with variable age and pollution controls over a one week period of sampling time. The study characterized PM contamination in typical Chinese steel cities with spatial significance. Four sampling cities were located at the Yangtze River Economic Zone (YREZ) which accounts for 20% of national Gross Domestic Product and is responsible for 1/3 of China's imports and exports. The cities selected in this study across southwest (Kunming), central (Wuhan) and east regions (Nanjing and Ningbo) covered developed and less developed areas in China. The sampling in this study was conducted during late April to early July when there was the shifting period from spring to summer in China. During this period, the weather was clear with moderate temperatures and humidity which was suitable for atmospheric particle sampling. The air pollution sources were relatively limited compared to wintertime which is typically dominated by coal combustion for heating purpose.

The results in this study showed that the PM emissions from iron and steelmaking process were associated with the age of the facilities, facility types and the capabilities of annual production of iron and steel. The trace element results showed that the small $PM_{1.1}$ particles were more contaminated than the intermodal and coarse fractions. As a result, $PM_{1.1}$ particles posed a negative impact on human health, specifically trace elements of As, Cr(VI) and Cd which presented carcinogenic risks via inhalation exposure. The Pb isotopic compositions showed that the local iron and steelmaking activities were the major source of PM contamination, followed by traffic and unidentified remote sources due to atmospheric movement. Results presented in this study implicate that sampling over a longer period is required to monitor air quality near industrial areas in order to confirm the impact of meteorological conditions on the transportation of atmospheric particles from the industrial sources.

## Supporting information

**S1 Fig. HYSPLIT backward trajectories at sites of Kunming (KM), Wuhan (WH), Nanjing (NJ), Ningbo (NB) and Ningbo Nottingham University (UN) during sampling periods in this study.**
(DOCX)

**S1 Table. Modelling parameters for PROMETHEE and GAIA analyse.**
(DOCX)

**S2 Table. Bioavailability analysis for trace elements.**
(DOCX)

## Acknowledgments

We acknowledge the assistance of Dr Shiva Prasad, Dr Andrew Evans, Dr Ping Di, Ms Fiona Zhang, Mr David Wo and Mr Richard Tea at the National Measurement Institute, North Ryde, Sydney, for their assistance with sample processing, data analysis and instrument determination.

## Author Contributions

**Conceptualization:** Xiaoteng Zhou.

**Software:** Xiaoteng Zhou.

**Supervision:** Vladimir Strezov, Yijiao Jiang.

**Writing – original draft:** Xiaoteng Zhou.

**Writing – review & editing:** Vladimir Strezov, Yijiao Jiang, Xiaoxia Yang, Tao Kan, Tim Evans.

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
