## [Decision Letter · Decision Letter 0]

13 Feb 2020

PONE-D-19-32739

Contamination identification, source apportionment and health risk assessment of trace elements at different fractions of atmospheric particles at iron and steelmaking areas in China

PLOS ONE

Dear Dr. zhou,

Thank you for submitting your manuscript to PLOS ONE. After careful consideration, we feel that it has merit but does not fully meet PLOS ONE’s publication criteria as it currently stands. Therefore, we invite you to submit a revised version of the manuscript that addresses the points raised during the review process.

We would appreciate receiving your revised manuscript by Mar 26 2020 11:59PM. To enhance the reproducibility of your results, we recommend that if applicable you deposit your laboratory protocols in protocols.io, where a protocol can be assigned its own identifier (DOI) such that it can be cited independently in the future. For instructions see: http://journals.plos.org/plosone/s/submission-guidelines#loc-laboratory-protocols

We look forward to receiving your revised manuscript.

Kind regards,

Bing Xue, Ph.D.

Academic Editor

PLOS ONE

Journal Requirements:

2. In your Methods section, please provide additional location information, including geographic coordinates for the data set if available.

3. We note that Figure 1 in your submission contains map and satellite images which may be copyrighted. All PLOS content is published under the Creative Commons Attribution License (CC BY 4.0), which means that the manuscript, images, and Supporting Information files will be freely available online, and any third party is permitted to access, download, copy, distribute, and use these materials in any way, even commercially, with proper attribution. For these reasons, we cannot publish previously copyrighted maps or satellite images created using proprietary data, such as Google software (Google Maps, Street View, and Earth). For more information, see our copyright guidelines: http://journals.plos.org/plosone/s/licenses-and-copyright.

You may seek permission from the original copyright holder of Figure 1 to publish the content specifically under the CC BY 4.0 license. 

If you are unable to obtain permission from the original copyright holder to publish these figures under the CC BY 4.0 license or if the copyright holder’s requirements are incompatible with the CC BY 4.0 license, please either i) remove the figure or ii) supply a replacement figure that complies with the CC BY 4.0 license. Please check copyright information on all replacement figures and update the figure caption with source information. If applicable, please specify in the figure caption text when a figure is similar but not identical to the original image and is therefore for illustrative purposes only.

Reviewers' comments:

Reviewer's Responses to Questions

**Comments to the Author**

1. Is the manuscript technically sound, and do the data support the conclusions?

Reviewer #1: Yes

Reviewer #2: Yes

Reviewer #3: Yes

2. Has the statistical analysis been performed appropriately and rigorously? 

Reviewer #1: Yes

Reviewer #2: Yes

Reviewer #3: Yes

3. Have the authors made all data underlying the findings in their manuscript fully available?

Reviewer #1: Yes

Reviewer #2: Yes

Reviewer #3: Yes

4. Is the manuscript presented in an intelligible fashion and written in standard English?

Reviewer #1: Yes

Reviewer #2: Yes

Reviewer #3: yes

5. Review Comments to the Author

Reviewer #1: The paper presents a comprehensive analysis in source apportionment of trace elements in atmospheric particles collected from four cities, China and estimates the associated health risks with the use of equations of hazard quotient (HQ) and cancer risk (CR), respectively. The paper could be accepted after addressing the following issues.

1. Line 92: Please define climate movement. What is climate movement?

2. Introductions: PM particles emitted from iron and steel industries posed negative human health not only through toxic trace metals, but also via polycylic aromatic hydrocarbons.

Liu et al., Environmental Pollution, 2019, 254, 113046.

Liu et al., Science of the Total Environment, 2020, 701, 134844.

Please add a brief introduction with the above references into the updated version of the paper.

3. Sampling information: How many samples do you collect for each site during the sampling period?

4. Data analysis: Please include backward trajectory analysis in the paper in order to testify the emissions from remote sources.

Reviewer #2: I have read the manuscript with great interest. I read it in detail and the truth is that the study is very good.

The way the problem, objectives and methodology are contextualize is appropriate. The results are clear and concise. The discussion is detailed and covers the explanation of all the results well. The conclusion highlights the most interesting results of the study.

English is clear and all sentences are well stated.

I consider that the munuscript is suitable to be published in Plos One without corrections. It is a very interesting work, which can be replicated in other industrialized cities in order to obtain very valuable information for the population and decision makers.

Reviewer #3: avoid long sentences.

6. PLOS authors have the option to publish the peer review history of their article (what does this mean?). If published, this will include your full peer review and any attached files.

Reviewer #1: No

Reviewer #2: No

Reviewer #3: No

---

## [Author Response · Author response to Decision Letter 0]

26 Feb 2020

Journal Requirements:

The formatting and file names were updated according to PLOS ONE’s style requirements. Thank you.

2. In your Methods section, please provide additional location information, including geographic coordinates for the data set if available.

The geographic coordinate for each location was added in the Method section. Please see Lines 119–121. Thank you for your comments to clear the manuscript. 

3. We note that Figure 1 in your submission contains map and satellite images which may be copyrighted. All PLOS content is published under the Creative Commons Attribution License (CC BY 4.0), which means that the manuscript, images, and Supporting Information files will be freely available online, and any third party is permitted to access, download, copy, distribute, and use these materials in any way, even commercially, with proper attribution. For these reasons, we cannot publish previously copyrighted maps or satellite images created using proprietary data, such as Google software (Google Maps, Street View, and Earth).

The satellite images included in the Figure 1 were created from the Google Maps. Due to the copyright consideration, these satellite images were removed, but its corresponding information was added as text in the Table 1. 

4. PLOS requires an ORCID iD for the corresponding author in Editorial Manager on papers submitted after December 6th, 2016.

The ORCID iD was updated in my PLOS ONE profile. Thank you.

5. Please include captions for your Supporting Information files at the end of your manuscript, and update any in-text citations to match accordingly.

The captions for the supporting information tables were added at the end of the manuscript. Thank you.

Reviewers' comments:

Reviewer #1: The paper presents a comprehensive analysis in source apportionment of trace elements in atmospheric particles collected from four cities, China and estimates the associated health risks with the use of equations of hazard quotient (HQ) and cancer risk (CR), respectively. The paper could be accepted after addressing the following issues.

1. Line 92: Please define climate movement. What is climate movement?

It was changed to ‘atmospheric movement’ to clarify this sentence. Please see Line 99. Thank you.

2. Introductions: PM particles emitted from iron and steel industries posed negative human health not only through toxic trace metals, but also via polycylic aromatic hydrocarbons.

Liu et al., Environmental Pollution, 2019, 254, 113046.

Liu et al., Science of the Total Environment, 2020, 701, 134844.

Please add a brief introduction with the above references into the updated version of the paper.

Both recommended references are nice work. However, this paper mainly focused on heavy metals rather than PAHs pollution, so there might be far from the topic to have an addition PAHs introduction in the current manuscript, but both papers were cities properly. Please see Line 51. 

3. Sampling information: How many samples do you collect for each site during the sampling period?

There were 200 samples collected and analysed in this study (8 samples per site per day). This detail was added on line 157. 

4. Data analysis: Please include backward trajectory analysis in the paper in order to testify the emissions from remote sources.

This backward trajectory analysis was added as S2 Figure in this study. 

Thank you for the reviewer’s comments to improve this manuscript. 

Reviewer #2: 

I have read the manuscript with great interest. I read it in detail and the truth is that the study is very good.

The way the problem, objectives and methodology are contextualize is appropriate. The results are clear and concise. The discussion is detailed and covers the explanation of all the results well. The conclusion highlights the most interesting results of the study.

English is clear and all sentences are well stated.

I consider that the munuscript is suitable to be published in Plos One without corrections. It is a very interesting work, which can be replicated in other industrialized cities in order to obtain very valuable information for the population and decision makers.

We highly appreciate the reviewer’s positive comments. Thank you again.

Reviewer #3: avoid long sentences.

The long sentence was found in Method section, and it has been broken down. Please see Lines 119–122.

---

## [Decision Letter · Decision Letter 1]

13 Mar 2020

Contamination identification, source apportionment and health risk assessment of trace elements at different fractions of atmospheric particles at iron and steelmaking areas in China

PONE-D-19-32739R1

Dear Dr. zhou,

We are pleased to inform you that your manuscript has been judged scientifically suitable for publication and will be formally accepted for publication once it complies with all outstanding technical requirements.

With kind regards,

Bing Xue, Ph.D.

Academic Editor

PLOS ONE

Additional Editor Comments (optional):

Reviewers' comments:

Reviewer's Responses to Questions

**Comments to the Author**

1. If the authors have adequately addressed your comments raised in a previous round of review and you feel that this manuscript is now acceptable for publication, you may indicate that here to bypass the “Comments to the Author” section, enter your conflict of interest statement in the “Confidential to Editor” section, and submit your "Accept" recommendation.

Reviewer #1: All comments have been addressed

Reviewer #2: All comments have been addressed

2. Is the manuscript technically sound, and do the data support the conclusions?

Reviewer #1: Yes

Reviewer #2: Yes

3. Has the statistical analysis been performed appropriately and rigorously? 

Reviewer #1: Yes

Reviewer #2: Yes

4. Have the authors made all data underlying the findings in their manuscript fully available?

Reviewer #1: Yes

Reviewer #2: Yes

5. Is the manuscript presented in an intelligible fashion and written in standard English?

Reviewer #1: Yes

Reviewer #2: Yes

6. Review Comments to the Author

Reviewer #1: The authors have addressed all the comments. Accept as it is, please.

The authors have addressed all the comments. Accept as it is, please.

Reviewer #2: All changes were incorporated by the authors, thus improving the manuscript. I consider that the work is apt to be published in PlosOne

7. PLOS authors have the option to publish the peer review history of their article (what does this mean?). If published, this will include your full peer review and any attached files.

Reviewer #1: No

Reviewer #2: No

---

## [Editor Report · Acceptance letter]

18 Mar 2020

PONE-D-19-32739R1 

Contamination identification, source apportionment and health risk assessment of trace elements at different fractions of atmospheric particles at iron and steelmaking areas in China 

Dear Dr. zhou:

I am pleased to inform you that your manuscript has been deemed suitable for publication in PLOS ONE. Congratulations! Your manuscript is now with our production department. 

With kind regards,

on behalf of

Professor Bing Xue 

Academic Editor

PLOS ONE